# Study on Filling Support Design and Ground Pressure Monitoring Scheme for Gob-Side Entry Retention by Roof Cutting and Pressure Relief in High-Gas Thin Coal Seam

**DOI:** 10.3390/ijerph19073913

**Published:** 2022-03-25

**Authors:** Hui Li, Haodong Zu, Kanglin Zhang, Jifa Qian

**Affiliations:** 1School of Urban Construction and Safety Engineering, Shanghai Institute of Technology, Shanghai 201418, China; lihui@sit.edu.cn (H.L.); 216132174@mail.sit.edu.cn (H.Z.); 216132166@mail.sit.edu.cn (K.Z.); 2College of Safety Science and Engineering, Henan Polytechnic University, Jiaozuo 454003, China; 3Henan Collaborative Innovation Center of Coal Work Safety and Clean High Efficiency Utilization, Jiaozuo 454003, China

**Keywords:** thin coal seam with high gas, gob-side entry retention, high-water material, backfill and support, strata behavior regularity

## Abstract

To ensure the successful application of roof cutting and pressure relief in the gob, to retain the roadway in a high-gas thin coal seam, by taking the 2109 working face of the Mingxin coal mine as the engineering background, this paper comprehensively analyzes and studies the key parameters of high-water material filling and support and the law of ground pressure behavior. The results show that the high-water material filling body has the characteristics of high strength, rapid resistance increase, strong flexibility and high strength in the later stage, which can meet the requirements for retaining roadway support along the goaf. On this basis, we determined that the water-cement ratio for a high-water material filling body is 1.5:1 and the filling length, height and width each time are 3.6 m, 2.2 m and 1.0 m, respectively. In addition, a ground-pressure monitoring scheme for retaining the roadway along the goaf is put forward and the results show that the displacement of the roof and floor and the deformation of the filling body are both within a reasonable range, which proves high-water material filling support can effectively ensure the stability and integrity of the roof of the gob, thus retaining the roadway in a high-gas thin coal seam.

## 1. Introduction

China is a country of massive coal production and consumption. In the report on China’s mineral resources issued by the Ministry of Natural Resources in October 2020, it was disclosed that coal accounted for 57.7% of the national energy consumption structure, and the output of raw coal increased by 4% over the previous year [1,2]. From this point of view, although clean energy is developing rapidly in China, coal will remain the main energy source in China for quite a long time [3]. However, with the rapid development of society, the demand for coal is also increasing, which is in obvious contradiction with the lack of renewable resources in China. Therefore, in the current environment, efficiently mining and utilizing coal resources with a high yield has become an urgent problem to be solved. 

Gob-side entry retaining has been widely used in domestic mines because it has the advantages of improving the resource recovery rate, reducing roadway excavation, easing the tension of mining replacement at the working face and prolonging mine service life [4,5]. Moreover, it can realize Y-type ventilation of the mine working face [6,7], solve the problem of gas accumulation in the upper corner of high-gas mine working faces, provide a guarantee of safe coal mining and obtain remarkable economic benefits. 

Domestic and foreign scholars conducted a lot of theoretical and practical research into the technical principle and application of retaining the roadway along the goaf. He et al. [8,9] proposed new technology for roof cutting, pressure relief and gob-side entry retention, which adopted pre-splitting of the blasting roof of the advanced working face and combined this with a double support form in and beside the roadway, to control the deformation of surrounding rock for entry retention. Kang et al. [10] used numerical simulations to analyze the deformation and stress distribution characteristics of the surrounding rock of deep gob-side retaining roadways, and put forward supporting principles and improvement suggestions for deep gob-side roadway retention by comparing on-site mine pressure-monitoring data. Ma et al. [11] studied the surrounding rock deformation law of gob-side roadways in a deep medium-thick coal seam under pressure relief by cutting the top, and put forward the technical requirements for surrounding rock control, which effectively controlled the surrounding rock deformation for roadway retention and achieved a good roadway formation effect. According to the activity characteristics of overlying strata, Chen [12] divided the surrounding rock deformation of the goaf, for roadway retention, into three stages, and obtained a calculation formula for the supporting filling body beside the roadway through a stable mechanical model of the key block structure. Li and Hua [13,14] divided the surrounding rock deformation into five stages by monitoring the surrounding rock change with roadway retention, and they summarized the surrounding rock deformation law of different stages. Xie et al. [15,16] used UDEC numerical software to simulate and analyze the distribution characteristics of the surrounding rock of the gob-side entry retention in fully mechanized caving, and carried out a numerical simulation calculation of the rock fracture and caving of the gob-retaining surrounding rock in the mining roadway, confirming the results for physical and mechanical parameters to ensure the stability of the surrounding rock. Zhu et al. [17,18] proposed that the surrounding rock deformation of gob-side entry retention is affected by many factors such as the surrounding rock structure, mechanical properties and ground stress, and they deduced a mathematical formula for the reliability of the support structure, putting forward corresponding filling support methods and measures. The research results of the above scholars have greatly promoted the development and application of gob-side entry retention in China’s modern coal industry and played an important role in promoting safe and efficient mining in China’s coal industry.

Roadway side filling support is key to the success of gob-side entry retention, and finding and using suitable filling materials has become key [19,20,21]. In the late 1970s, British coal mines successively developed three kinds of bag filling materials: fly ash + cement + suspension agent, cement + bentonite and bauxite cement + calcium carbonate + bentonite. In terms of support strength, the third mixed material was significantly better than the first two. Subsequently, based on these three materials, the British coal mines developed a filling material with a water distribution ratio of only 90%, which has since been widely used in the British coal mining industry [22,23]. German coal mines have mainly studied synthetic gypsum material, fly ash + Portland cement mixed material and gangue + gel coagulant material, and achieved good results in gob-side roadway-retaining support [24]. As the first country in the world to undertake gob-side entry retention, Poland initially used water and sand to form the side-entry retaining wall, and later used gangue belt support, dense pillars, concrete walls and filling materials for side-entry support, which achieved good results and was successfully applied in coal mines with complex geological conditions [25]. In China, most of the early studies used woodpile, gangue and dense pillars for roadway side-filling support. However, these methods consume great manpower, material and financial resources, increase the resistance slowly and cannot seal the goaf [26,27,28,29]. With the development of gob-side entry retention, scholars have successively developed paste materials and high-water materials for roadway side-filling support. These two materials cannot only provide strong support resistance, to effectively control the deformation of the surrounding rock in the reserved roadway, but also have a good effect on the closed goaf. In addition, these two materials well meet the requirements of national energy policies such as green mining and sustainable development, meaning they have been widely used in the Chinese coal mining industry.

Two new roadway side-support technologies, paste filling material and high-water filling material, have been researched and developed successively. These two methods can effectively control the deformation of the surrounding rock in the entry, and have good tightness to the goaf and strong support resistance [30,31,32].

Although scholars at home and abroad have done a lot of systematic research, the complex mining and geological conditions in China’s coal mines hinder the application of gob-side entry retention and roadway side-filling support. For mines with high-gas and thin coal seam geological conditions, the factors of stress concentration, high gas emissions and great mining difficulty increase the challenge of gob-side entry retention and roadway side-filling support. In addition, the above research mostly focused on the technical principle of retaining the roadway along the goaf and the performance of roadway side-filling materials, but there is little research on how to monitor the ground pressure after the application of roadway retention along the goaf. 

Therefore, in view of the above problems, this paper takes the 2109 working face of the Qinyuan Mingxin coal mine as the research object and analyzes the technical principle of roadway side-filling support at a high-gas thin coal seam. We design and determine the key parameters and implementation scheme for roadway side-filling support technology with a high-water material. Finally, by using a variety of monitoring methods, the ground pressure behavior law of the roadway retained along the goaf after introducing the roadway side-filling support is monitored and analyzed, and the effect of the roadway side-filling support on the 2109 working face is investigated via the ground-pressure monitoring results.

The research results of this paper not only provide a comprehensive scientific reference for the future application of high-water-material gob-retaining-roadway filling-support technology in high-gas thin seam mines but also provide an effective ground-pressure monitoring scheme and guidance for similar mines, which is of great significance to ensure the safety of mining production work.

## 2. Engineering Geological Conditions

### 2.1. Mine Overview

Qinyuan Mingxin Coal Mine Co., Ltd. is located in the south of Qinyuan County, Changzhi City, Shanxi Province, China. The mine field area is 5.9 km^2^, the retained coal reserves are 25.30 million tons, the recoverable reserves are 19.25 million tons and the service life is 16.5 years. The mine field is generally a wide and gentle monoclinal structure with a nearly north-south strike, and the dip angle of the coal seam is 3–7°. The currently mined 2# coal seam, with an average thickness of 1.07 m, belongs to high-quality main coking coal, with recoverable reserves of 6.54 million tons and a service life of 5.6 years. The strike longwall method is adopted for coal mining, and the full caving method is used to manage the roof.

Qinyuan Mingxin Coal Mine is a high-gas mine with an absolute gas emission of 22.23 m^3^/min and a relative emission of 11.74 m^3^/t. The spontaneous combustion tendency of the coal seam is grade III, which means it is a coal seam that will not easily spontaneously combust, but the coal dust is explosive.

### 2.2. Working Surface Layout

The test site of the project is the 2109 working face, which is located in the east wing of the first mining area. It is a monoclinic structure and belongs to the 2# coal seam. The 2109 working face is arranged in a rectangular shape, with an average strike length of 570 m, an inclined length of 150 m, an area of 108,500 m^2^ and an industrial reserve of 160,380 tons. The roof is mostly mudstone or fine-grained sandstone, and the thickness is generally 1.7–3.2 m, belonging to the medium stable type. The floor is mostly mudstone or siltstone with a thickness of 2.2–3.5 m, belonging to the unstable type. The position of the gob-side entry retention is the 2109 auxiliary transportation trough, and the geometric dimensions of the working face transportation trough are as follows: the cross-section is rectangular, the net width × net height is 4.2 m × 2.4 m and the section area is 10.08 m^2^. The cutout size of the working face is as follows: the section is rectangular, the net width × net height is 5.2 m × 2.4 m and the section area is 12.48 m^2^. The cut hole is first driven along the groove section and then expanded to the design section after penetration. The layout of the working face and the position of the gob-side entry retention (marked with red dotted line) are shown in Figure 1.

The direction of the 2109 working face is northeast-southwest, and the inclination is southeast. The coal seam is high-quality main coking coal, relatively stable, mainly bright coal, with a small amount of mirror coal and dark coal included, which belong to semi-bright briquette. The thickness of the coal seam is 0.75–1.93 m, and the average thickness is 1.07 m. Generally, it does not contain gangue, and the structure is simple. The dip angle of the coal seam is 3–7°, and the whole can be regarded as a near-horizontal coal seam. The coal seam is well-developed with cracks and belongs to a stable mineable coal seam spanning a wider area. According to the geological report, it is predicted that the coal seam will gradually thicken in the due west direction of the roadway, with local fluctuations, and the roadway fluctuates and changes along the dip angle of the coal seam. Figure 2 shows the rock pillar characteristics and physical-mechanical indexes of each coal stratum in the 2109 auxiliary roadway.

## 3. Technical Principle of Filling Support of Gob-Side Entry Retention in High-Gas Thin Coal Seam

According to the exploration of the potential for side-entry filling in gob-retaining roadways, the roof deformation of gob-retaining roadways can be divided into three stages, namely, the primary mining influence stage, the retaining roadway stability stage and the secondary mining advanced influence stage [33,34]. The deformation characteristics and stress conditions of the filling support body beside the gob-retaining roadway in each stage are as follows:(1)Primary mining influence stage: due to the mining of the previous working face, the roof collapses freely due to its excessive exposed length and automatically fills the goaf, resulting in layer separation. When the working face advances the distance of two-periodic weighting, the roof gradually breaks, rotates and sinks. The pressure on the roadway side-filling body directly below the key block increases, and the deformation also increases. At this time, if the support resistance of the filling body is insufficient or has no deformation capacity, the filling body may be damaged. This stage is the most difficult stage of retaining a roadway along a goaf.(2)Stability stage of retaining a roadway along a goaf: with the continuous advancement of the working face, the “hinge” structure between the key blocks of the overlying strata will not change greatly. The deformation speed of the surrounding rock of the roadway reserved along the goaf and the lateral support pressure of the goaf tend to be stable, and the pressure on the filling body beside the roadway remains basically unchanged. However, with the roof sinking, the strength of the filling body will change. If the strength of the filling body is insufficient at this time, it can also cause instability failure.(3)Secondary mining advance influence stage: when the mining work of the second working face is gradually approaching, the gob-side entry will be in the superposition area of the support pressure of the first and second working faces, and the support pressure will reach the maximum near the working face. At this time, the pressure on the filling support increases rapidly, and severe deformation may occur. However, the gob-side entry is mainly used for return air in the mining process of the second working face so it only needs to meet the ventilation conditions without reinforcement and support.

Based on the stress and deformation characteristics of the filling body in the above three stages, the supporting mechanism of the goaf-retaining filling body can be obtained as follows:(a)The backfill body must have a certain strength and the characteristics of rapid resistance increase in the initial stage. At the initial stage of advancing the working face, to ensure the integrity of the roof within a certain range on the upper side of the goaf and ensure the joint movement of the upper and lower strata, a filling body with certain strength should be built in the roadway, in time, after the advancing of the working face, to prevent the occurrence of separation and damage. The rapid resistance increase of the filling body is to have enough roof-cutting resistance to cut off a certain height of roof before the roof breaks and sinks, to reduce the bearing load of the roadway side-support, ensure that the key blocks contact the falling gangue in the goaf earlier and form a stable structure.(b)The filling body should have a certain amount of shrinkage. Due to the deadweight pressure of overlying strata and the rotation and subsidence of key blocks, it is difficult for the backfill to prevent its movement track. Therefore, in the stage affected by mining operation, the backfill should also have good deformation adaptability. It can adapt to the movement of the roof through appropriate retraction, to reduce the pressure and ensure that the backfill will not be destroyed. At the same time, moderate compression can make the direct roof rotate and sink by coming into timely contact with the gangue of the goaf, thus forming a stable structure together with the key block.(c)The filling body should have high post-strength. After the goaf-retaining roadway is in the stable stage, the filling body beside the roadway needs strong later strength to ensure that the roadway will not be seriously deformed or become unstable, especially in the advanced influence stage of the second working face, when it is key to ensure that the roadway meets the ventilation demand.

Therefore, in combination with the above technical principle of gob-side entry-retaining filling support in a high-gas thin coal seam, the project used high-water materials to form a roadway backfill wall by side-pouring near the goaf, for roadway retention after advancing the working face. The roadway side-filling wall produced roadway side-support resistance, which promoted the roof of the goaf to collapse along the pre-split cutting joint in time, and the cut rock blocks of the immediate roof and basic roof could fill the goaf beside the roadway so that the overlying strata of the roadway formed a short boom structure, as shown in Figure 3.

## 4. Key Parameters of Backfill and Support for Gob-Side Entry Retention in High-Gas Thin Coal Seams

### 4.1. Selection and Performance of Filling Materials Beside Gob-Side Entry to Retain Entry

Although the filling strength of paste material is high, its filling process is complex, the cost is high and the conveying distance is small. By contrast, the use of high-water material filling cannot only provide large initial support resistance and a fast resistance increase speed but also a high degree of mechanization and easy transportation over long distances, making it highly suitable for large-scale gob-side entry retention. Therefore, this study used high-water materials to fill and support gob-side entry retention at the 2109 working face of the Qinyuan Mingxin Coal Mine.

High-water material is an inorganic mixture that can rapidly coagulate under a high water-cement ratio. Its components are solid powders A and B, as shown in Figure 4a. The filling body can be obtained by uniformly mixing and solidifying the slurry formed, by separately adding water to and stirring solid materials A and B, as shown in Figure 4b. During on-site construction, A and B slurries are transported to the filling site through two sets of independent pipelines, and then they are mixed through the mixing pipeline and poured into the filling bag. A slurry and B slurry are mixed and then harden quickly. The performance of high-water materials is mainly related to the water-cement ratio. To determine the optimal high-water material ratio to meet the engineering needs, by using the MTS815 electro-hydraulic servo rock mechanics testing machine (MTS, USA), as shown in Figure 4c, the high water-cement ratio conditions—i.e., the uniaxial compressive strength of the water material filling body, as well as the resistance increase rate and deformation performance of the high-water filling material—were tested, and the results are shown in Figure 4d–f.

Figure 4d shows the relationship between the uniaxial compressive strength of high-water materials and the water-cement ratio. It can be seen that the uniaxial compressive strength of the high-water filling body is inversely proportional to the water-cement ratio, that is, the smaller the water-cement ratio (the more high-water materials required for filling per unit volume), the higher the strength of the condensate. Conversely, the larger the water-cement ratio (the less high-water material required per unit volume of filling), the lower the condensate strength. Using the first-order exponential function to fit it, the expression between the uniaxial compressive strength and the water-cement ratio can be obtained as: $y=4.5+123.6e^{\left(-2x\right)}$, and the fitting degree is 0.99. During on-site construction, the compressive strength of the backfill can be changed by adjusting the water-cement ratio to meet the engineering needs.

In addition to having a certain compressive strength, a faster resistance increase rate of the backfill body along the gob is also important for roadside support of the gob-side entry retention. The two components A and B of the high-water filling material selected for the current study can be initially coagulated after adding water and mixing for 20 min. The specific variation of the strength of the high-water backfill with time is shown in Figure 4e (the water-cement ratio of the backfill in the figure is 1.5:1). It can be seen that the compressive strength of the high-water filling material can reach 2.1 MPa after 2 h, 5.6 MPa after 1 d of mixing, 10.36 MPa after 7 d and 10.82 MPa after 28 d. In other words, it only takes one day for the high-water filling material to reach the final strength of more than 50%, and it only takes seven days to reach the final strength of more than 90%, which shows that the high-water filling material can harden rapidly and achieve a certain strength sufficient to generate support resistance.

Figure 4f shows the total stress-strain curve of the high-water material backfill with a water-cement ratio of 1.5:1 and an age of 7 d. It can be seen that under this condition, the peak compressive strength of the high-water filling material is 10.36 MPa, and the corresponding strain is 6.5%. When the strain continues to increase to 10%, its compressive strength can still maintain more than 65% of the peak strength, and when the strain continues to increase to 18%, the residual compressive strength of the filling is more than 59% of the peak strength. This shows that the high-water material has outstanding plastic characteristics. After the load reaches the peak strength, the high-water material does not immediately and completely fail; instead, as the strain further increases, the bearing capacity decreases slowly. The rate of decline is much smaller than that of general concrete and rock material. Therefore, in the field, during application, the high-water filling material roadside support body can produce large plastic deformation under pressure and maintain a high residual strength in the later stage so that the roadway will not be seriously deformed and will meet subsequent ventilation requirements.

### 4.2. Calculation of the Support Resistance of the Filling Body Beside the Gob-Side Entry Retention

Determining the reasonable support resistance for gob-side entry retention is a process of finding the critical support resistance. Once this is determined, the support resistance of gob-side entry retention may also be determined. Combined with the actual situation of the 2109 working face in the Qinyuan Mingxin Coal Mine, based on the roof load-strip separation method model (as shown in Figure 5a), a long strip of one unit width was taken near the ABCD area (as shown in Figure 5b) for analysis, and we further obtained the calculation model of the backfill support resistance of the gob-side entry retention, as shown in Figure 5c.

Assuming that the uniform load on the top plate is *Q*, the load on the long strip in Figure 5b is distributed at both ends. The support resistance for gob-side entry retention is represented by the concentrated load *P* of the support at the side of the roadway, and the support resistance of the support in the roadway is neglected. According to limit analysis theory, when the rock is broken, the limit bending moment at the breaking point is *M*, as shown in Figure 5c. The moment balance method is used to analyze the sections *A*’*B* and *AA*’ in Figure 5c, and the resistance of the first layer above the gob-side entry retention can be obtained, as shown in Equation (1):(1)$$\left\{\begin{array}{l} F_{A’}-q_{1}L_{1}=0\\ 2M_{P1}-q_{1}L_{{}_{1}}^{2}/2=0\\ M_{A1}-M_{P1}-q_{1}a^{2}/2-F_{A’1}Q+P_{1}a=0 \end{array}\right.$$
where $F_{A’}$ is the downward shear force generated by the breaking of rock stratum at point *A*’ of the first roof, $q_{1}$ is the gravitational concentration of rock stratum of the first roof, $L_{1}$ is the rock failure characteristic size of the first roof, $M_{P1}$ is the ultimate bending moment of rock stratum of the first roof, $M_{A1}$ is the anti-bending moment of rock stratum of the first roof, a is the roadway maintenance width and $P_{1}$ is the cutting resistance.

Simultaneously, we solve:(2)$$P_{1}a=M_{P1}-M_{A1}+q_{1}a^{2}/2+q_{1}L_{1}a$$
where $q_{1}=\gamma_{1}h_{1}$ and $M_{A1}=M_{P1}$ (under extreme conditions).

Considering the mining impact, the mining impact coefficient *k* is substituted into Equation (2) to get:(3)$$P_{1}=k\left\{\gamma_{1}h_{1}a/2+\gamma_{1}h_{1}L_{1}\right\}=k\left\{\gamma_{1}h_{1}\left(a/2+L_{1}\right)\right\}$$
where $\gamma_{1}$ is the rock bulk density of the first roof and $h_{1}$ is the depth of stratum of the first roof. 

Different from the roof cutting resistance of the first layer, which is mainly provided by artificial support, the roof cutting resistance required by the rock layers above the second layer of gob-side entry retention is formed by the joint action of artificial support and the residual boundary of the collapsed rock layer. The force distribution is shown in Figure 5d. Similarly, analyzing the *A*’*B* segment and *AA*’ segment, we can get:(4)$$\left\{\begin{array}{l} F_{A’2}=q_{2}L_{2}=\gamma_{2}h_{2}L_{2}\\ M_{A}{}_{1}+M_{A2}+P_{2}a-M_{P2}-\gamma_{2}h_{2}{\left(a+h_{1}tg\alpha_{1}\right)}^{2}/2\\ -\gamma_{1}h_{1}a^{2}/2-F_{A’2}\left(a+h_{1}tg\alpha_{1}\right)-F_{A’1}a=0 \end{array}\right.$$
where $\alpha_{1}$ is the rock breaking angle. 

Substitute the mining influence coefficient k into Equation (4), we solve and simplify it to get:(5)$$P_{2}a=k\left\{\sum_{i=1}^{2}\gamma_{i}h_{i}{\left(a+\sum_{j=0}^{i-1}h_{j}tga_{j}\right)}^{2}/2+\sum_{i=1}^{2}F_{A’i}\left(a+\sum_{j=0}^{i-1}h_{j}tga_{j}\right)+M_{P2}-\sum_{i=1}^{2}M_{A}{}_{i}\right\}$$

In the same way, the general formula for calculating the roof cutting resistance of the gob-side entry-retaining support for the *m*-th layer of rock can be obtained as follows:(6)$$P_{m}a=k\left\{\sum_{i=1}^{m}\gamma_{i}h_{i}{\left(a+\sum_{j=0}^{i-1}h_{j}tga_{j}\right)}^{2}/2+\sum_{i=1}^{m}F_{A’i}\left(a+\sum_{j=0}^{i-1}h_{j}tga_{j}\right)+M_{Pm}-\sum_{i=1}^{m}M_{A}{}_{i}\right\}$$
where *i* is the rock stratum of roof *I*, *j* is the rock stratum of roof *j* and m is the limit number of strata in the upper part of goaf, which can be obtained by dividing the total thickness of strata by the average thickness.

Assuming that the layered caving thickness of each rock layer, breaking angle, bulk density and ultimate breaking moment are the same, that is, when $h_{i}=h$, $\alpha_{i}=\alpha$, $\gamma_{i}=\gamma$ and $M_{Pi}=M_{P}$, Equation (6) can be simplified as:(7)$$P_{m}a=k\left\{\begin{array}{l} \frac{1}{2}\gamma h\left[ma^{2}+ahtg\alpha m\left(m-1\right)+h^{2}tg^{2}\alpha m\left(m-1\right)\left(2m-1\right)/6\right]\\ +a\sum_{i=1}^{m}F_{A’i}+htg\alpha \sum_{i=1}^{m}F_{A’i}\left(i-1\right)+M_{Pm}-mM_{A} \end{array}\right\}$$

When each layer within *m* − 1 is cut off and loses contact with the residual boundary, then $\sum_{i=1}^{m-1}F_{A’i}=0$ and the maximum value of $F_{A’m}$ is $\gamma hL_{m}$. Substituting this into Equation (7), the calculation formula of the support resistance of the gob-side roadway can be obtained as:(8)$$P_{m}a=k\left\{\begin{array}{l} \frac{1}{2}m\gamma h\left[a^{2}+ahtg\alpha \left(m-1\right)+h^{2}tg^{2}\alpha \left(m-1\right)\left(2m-1\right)/6\right]\\ +\gamma hL_{m}\left[a+htg\alpha \left(m-1\right)\right]+M_{Pm}-mM_{A} \end{array}\right\}$$
where $M_{Pm}$ is the ultimate bending moment of the *m*-th floor, $M_{Pm}=\frac{1}{6}R_{t}h^{2}$, $R_{t}$ is the tensile strength of stratum of the *m*-th floor and $M_{A}$ is the bending moment of each stratum at point *A*. 

Combined with the geological conditions of the 2109 working face, the mining conditions of adjacent working faces and the properties of the rock mass, the specific geomechanical parameters (as shown in Table 1) are substituted into Equation (8), and the support resistance per meter of roadway reserved along the goaf is P = 7.23 MN/m, that is, the supporting strength of the gob-side entry retention shall not be lower than 7.3 MPa, so that the surrounding rock of the gob-side entry retention can be kept stable.

### 4.3. Determination of Key Parameters of Backfill and Support Body Beside Gob-Side Entry Retention

(1)Size of filling body beside gob-side retaining entry

The size of the backfill beside the gob-side retaining entry refers to the length, height and width of the backfill formed by each filling. Among them, the length mainly depends on the working face’s propellant-filling operation system. During on-site construction, the 2109 working face is filled once a day, and a one-shift operation system is adopted. The daily advancing length is 3.6 m, and the length of the filling body formed by each filling is 3.6 m. The height of the filling body is theoretically the mining height, while the actual filling height is the distance between the roof and floor of the filling area. According to the layout of the 2109 working face, the height of the backfill is 2.2 m based on the average mining height in the workshop. Under the premise that the length and height of the filling body and its required supporting strength have been determined, the width of the filling body becomes the main factor affecting its strength. According to the analysis results of Section 4.1 on the performance of high-water filling materials, a water-cement ratio of 1.5:1 is selected to mix high-water materials to form a filling body, and the relationship between the strength and width of the filling body under this condition is shown in Figure 6.

When the water-cement ratio is 1.5:1, the average strength of the high-water-material roadside backfill is 8 MPa, and the strength of the high-water-material backfill gradually decreases with the increase of the width. According to the calculation results of the support resistance of the backfill beside the gob-side entry retention in Section 4.2, the support strength of the backfill beside the roadway should not be lower than 7.3 MPa. Considering the strength of the backfill body and the cost of backfilling, this paper selects gob-side entry-retaining high-water material for a backfill body width of 1.0 m.

Therefore, when filling the 2109 working face of the Mingxin Coal Mine with high-water materials beside the gob-side entryway, the construction shall be carried out according to the filling length of 3.6 m, height of 2.2 m and width of 1.0 m each time, to ensure that sufficient support is provided. In this way, the gob-side entry retention helps the surrounding rock to remain stable.

(2)Location of the backfill beside the gob-side retaining entry

According to the experience of backfilling and support beside a gob-side retaining entry [35,36,37], there are usually three types of layout schemes for the location of the filling body beside the gob-side retaining entry: one is when the width of the roadway is sufficient, meaning the wall is located in the roadway; the second is when the width of the roadway is too small (when the requirements are not met, the wall is located on the side of the goaf); third, when the width of the roadway is small and needs to be properly widened, part of the wall is located on the side of the gob, and another part of the wall is located in the roadway. Considering the engineering conditions of the 2109 working face of the Mingxin Coal Mine, this paper selects the first scheme to arrange the backfill beside the gob-side entry retaining. The backfill and roadway positions formed by high-water materials are shown in Figure 7a. In addition, when carrying out the filling work beside the gob-side entry retention, it is necessary to set up a single double-row of dense gangue retaining pillars on the side of the support end of the mining side (as shown in Figure 7b), and double-layer metal mesh is hung on the roof to prevent debris from falling. The gangue falls, providing a safe space for filling operations. Two hours after completing the filling operation, the gangue retaining pillar shall be moved forward or removed with the advance of the working face.

## 5. Monitoring and Developing Law of Gob to Retain Roadway Pressure in High-Gas Thin Coal Seam

For the period of the filling-body supporting effect on the implementation of the mine, pressure monitoring is to ensure and test for the success of the important and most effective means. Through the analysis of real-time monitoring data can be timely follow up master along the empty left lane roadway surrounding rock deformation and damage in the process, to the left lane using existing filling body support analysis evaluation on the safety and reliability.

### 5.1. Content and Scheme of Mine Pressure Monitoring for Goaf to Retain Roadway

The field monitoring test is used to monitor the change of mine pressure in the gob-side retaining entry at the 2109 working face, and the validity and reliability of the proposed support scheme are verified by engineering. The specific monitoring content and program are as follows:

#### 5.1.1. Roadway Surface Displacement Monitoring

This mainly monitors the surface movement of the roof, floor, roadway and backfill wall.

For the monitoring of the surface movement of the roof and floor and the two sides of the roadway, an observation point is set every 20 m outward from the cutting position of the 2109 working face. The observation point adopts the cross-observation method for surface displacement monitoring, and the layout of the observation point is shown in Figure 8. Take point O as the base point, set point D on the bottom plate with a wooden pile as the monitoring point and make it vertically penetrate the bottom plate. The bolt outcrop of the roadway section is taken as the monitoring point for both sides and the roof. The whole monitoring process is mainly completed by measuring tools such as a measuring gun and steel tape. During the measurement, the distance between two points AB and CD shall be measured with measuring tools such as a measuring gun and steel tape. The measurement accuracy is required to be ±1 mm. Each observation point was measured twice, and a plane cartesian coordinate system was established according to data changes of the measured distance at each point, while the deformation of each measuring point was calculated according to the monitoring data. The displacement data for the roof, floor and roadway sides were measured and recorded once a day.

For the monitoring of the surface movement of the filled wall, the crossing observation method was also used to arrange the measuring points on the surface of the filled wall, as shown in Figure 9. The specific method was to arrange the base point O on the centerline of each filling body, and measure the distance between points OA and OB, i.e., the longitudinal displacement of the filling body, with measuring tools such as a measuring gun and steel tape. Since there was no interval between adjacent filling bodies, the distance between C and D was basically unchanged. So, it was no longer measured and recorded. The initial value was recorded after the completion of the filling wall, and then the surface displacement data of the filling body was recorded once a day.

#### 5.1.2. Roadway Roof Deformation Monitoring

The main deformation of the roadway roof was bending subsidence, layer separation and roof falling. To well ooze Ming auxiliary transport of coal mine 2109 working face of gateway for a long period of roof deformation and damage monitoring, timely to better understand the changes in the deep strata of roof of roadway near the starting cut 270 m by the direction of the line location, 2109 auxiliary transport gateway every 50 m to set up a monitoring station. A set of LBY-3 roof separation monitors was installed in each monitoring station, 700 mm away from the filling wall, as shown in Figure 10. Among them, the installation steps of the roof separation layer monitor were as follows: first, we drilled holes φ 28 mm and 10.5 m deep at the corresponding position, then pushed the deep base point anchor into the hole until the designated position with anchor cable. Then, we pulled the anchor cable to confirm that the anchor had been fixed, and pushed the shallow base-point anchor into the designated position in the hole, pulling the wire rope to make sure the anchor was secured. Finally, the white PVC sleeve of the layer separator was pushed into the hole for fixation; its lower end was close to the top plate. We recorded the initial readings in the shallow and deep parts. The data on roof separation deformation were measured and recorded once a day.

#### 5.1.3. Anchor Cable Stress Monitoring

The change of anchor cable stress can reflect the stress of surrounding rock and the effect and reliability of support. The anchor cable working load was monitored by installing an anchor cable dynamometer between the anchor cable tray and the nut (as shown in Figure 11). When we entered the roof pressure, pressure box cover force, pressure box lithium lubricating oil pressure and pressure gauge tubing into the force table and displayed the force value, the size of the force value was proportional to the pressure box force. The stress variation of the surrounding rock was reflected by the stress variation of each monitoring point when the working face was constantly mining during the gob-retaining period. According to the actual situation, anchor cable dynamometers were installed from 180 m outward from the cutting hole along the goaf retaining roadway, with a group of dynamometers installed every 50 m. Two dynamometers were installed in a group on two anchor cables in the middle of the roadway and at the side of the goaf, as shown in Figure 11. We recorded the initial value of the dynamometer after the stress gauge was installed, taking an anchor cable dynamometer pressure reading once a day.

### 5.2. Analysis of Ore Pressure Appearance Law of Goaf to Retain Roadway

#### 5.2.1. Displacement Analysis of Roof, Floor and Two Sides of Roadway

Through calculation and analysis of the measured data, two groups of typical surface displacement variations of the roof, floor and two sides of roadway were obtained, as shown in Figure 12 and Figure 13. As can be seen:a.When the surrounding rock tended to be stable, the cumulative movements of the roof and floor of measuring points No.1 and No.2 were 243 mm and 223 mm, respectively, and the average displacement was 233 mm. The two sides of measuring points No.1 and No.2 were 203 mm and 168 mm, respectively, and the average displacement was 185.5 mm.b.In terms of displacement rate, the top and bottom moving velocity of No.1 was generally larger than that of No.2. The maximum moving velocities of No.1 and No.2 were 13.16 mm·d^−1^ and 12.63 mm·d^−1^, respectively. The approaching velocity of the two sides of the roadway at measuring point No.1 was also greater than that at measuring point No.2, the maximum was 10.53 mm·d^−1^, which was obviously greater than that at measuring point No.2 (8.64 mm·d^−1^).c.Within 20 m behind the working face (<5 d), the roof activity was not obvious, and the movement of the roof, floor and two sides was small. In the range of 20 m–40 m (5 d~8 d) behind the working face, the displacement of the top, bottom plate and two sides began to increase slowly. At the rear of the working face, within the range of 40 m–100 m (8 d~20 d), the roof activity was violent, and the relative movement of the roof, floor and two sides increased rapidly. In the range of 100 m to 150 m (20 d to 30 d) behind the working face, the movement of surrounding rock slowed down, and the increase rate of the roof, floor and lateral displacement decreased gradually. After the range of 150 m (>30 d), the surrounding rock activities of both sides and the roof and floor of the roadway entered a stable stage.d.The roof, floor and the two sides were within the reasonable and controllable deformation range, indicating that the filling and support effect of high-water material beside the gob-side retaining roadway met the expected requirements and could meet the engineering requirements.

#### 5.2.2. Deformation Analysis of Backfill Body

By analyzing and organizing the observed data measured at the filling-body measurement points, the deformation curve of the filling body was drawn, as shown in Figure 14. 

It follows that:During the whole gob-side entry-retention process, the deformation law of the filling body in the gob-side entry retention of the 2109 working face can be divided into four stages: (I) In the range of 0–30 m (<6 d) behind the working face, the rotary subsidence of the rock beam in the roof of the working face is small, and the pressure acting on the filling body beside the roadway is small. So, the deformation rate of the filling body is relatively slow. Moreover, due to the small gap between the initial filling wall and the roof of the retaining roadway, the displacement of the initial filling wall is smaller than that of the top and bottom plate. (II) In the range of 30–100 m behind the working face (6 d~20 d), the roof rock beam settlement is severe, and the load on the filling body increases rapidly, while the deformation rate of the filling body is large. (III) In the range of 100–140 m (20 d~30 d) behind the working face, the surrounding rock movement of the roof becomes slow, the load on the filling body increases slowly, and the deformation rate of the filling body decreases, while the deformation is still increasing. (IV) After 140 m (>30 d) behind the working face, the deformation of surrounding rock in the working face tends to be stable. The filling body beside the roadway supports the roof without being crushed, and the deformation rate is about zero.According to the above monitoring results, the final deformation of the filling body is small, about 140 mm, and there is no instability failure of the filling body. Due to its compressibility, the filling body can still maintain its due supporting performance after deformation, which plays an effective role in supporting the roof rock beams in the roadway, and can meet the safety requirements of roadway retention in mining.

#### 5.2.3. Deformation Analysis of Roadway Roof

According to the data results monitored at the measuring points of the roadway roof separation layer, the variation rule of roof displacement at measuring points No.1 and No.2 was obtained, as shown in Figure 15 and Figure 16.

The analysis results are as follows:During the monitoring period, the accumulated separations of the roof depth base at points No.1 and No.2 were 36 mm and 32 mm, respectively, and the average accumulated separation amount was 34 mm. The accumulated amounts of layer separation at the shallow base point were 30 mm and 27 mm, respectively, and the average accumulated amount of layer separation was 28.5 mm. The maximum displacement velocity of the two measuring points was 2.5 mm·d^−1^ and the minimum displacement velocity was 0.5 mm·d^−1^ at the deep base point; the maximum displacement velocity of the shallow base point was 2.25 mm·d^−1^ and the minimum displacement velocity was 0.38 mm·d^−1^.No.1, No.2 and deep shallow bp basis points of maximum cumulative abscission rate is no greater than 40 mm, and two measuring point deep basis points and the magnitude of the abscission layer between shallow basis points difference is no more than 10 mm, anchor cable and the pack on supporting played a good effect in the process of roof subsidence, effectively controlling the deformation of the surrounding rock, roof give full play to the since the carrying capacity of the rock. As such, the expected outcome of roadway side-support installation was achieved.With the continuous advance of the working face, the changes of monitoring data at each measuring point tended to be stable. The surrounding rock structure was stable and change gradual, and the amounts of stratification at shallow and deep base points tended to be stable at about 28 d and 30 d of entry retention.

#### 5.2.4. Anchor Cable Stress Analysis

Two anchor cables in the middle of the roadway and the side of the goaf were selected for monitoring at measuring points No.1 and No.2. Through sorting and analysis of the monitoring data, the typical stress variation rules of anchor cables at measuring points No.1 and No.2 were obtained, as shown in Figure 17 and Figure 18. 

We determined that:Over time, the stress on the anchor cable increased in the middle of the roadway and at the goaf side. The cumulative stress levels on the anchor cable in the middle of the roadway at measurement points No.1 and No.2 were 27 MPa and 24 MPa, respectively, and the cumulative stress levels on the anchor cable at the goaf side were 22 MPa and 20 MPa, respectively. Due to the influence of continuous mining at the working face, the subsidence of the goaf side-roof led to the transfer of the roof stress peak to the top rock layer of the solid coal side, and then the stress of the anchor cable in the middle of the roadway at the measurement points No.1 and No.2 was greater than that of the goaf side.The maximum growth rates of the anchor cable stress in the middle of the roadway at measurement points No.1 and No.2 were 1.27 MPa·d^−1^ and 1.28 MPa·d^−1^, respectively, and the maximum growth rates of the anchor cable stress at the goaf side were 1.11 MPa·d^−1^ and 1.14 MPa·d^−1^, respectively. Regardless of whether in the middle of the roadway or at the side of the goaf, the maximum growth rate of the anchor cable pressure at measuring points No.1 and No.2 was obtained within 10 d of mining advance, that is, 50 m from the working face, indicating that the deformation of surrounding rock by the roadway is large within this range.The stress on the anchor cable at each measuring point tended to be stable, and the stress activity of the surrounding rock tended to be stable after mining for about 28 days. The stress increment of the anchor cable in the middle of roadway and goaf was relatively small and in a controllable range, reflecting the good supporting effect of high-water material support.

## 6. Conclusions

Taking the 2109 working face of the Mingxin Qinyuan coal mine as the experimental area, this paper carried out research on the high-water filling support and ground-pressure monitoring of a gob-retaining roadway in s high-gas thin coal seam. The main conclusions are as follows: (1)According to the stress characteristics of the backfill body at different stages, the support mechanism of the backfill body for gob-side entry retention was obtained as follows: (a) the backfill body must have sufficient strength at the initial stage and can realize rapid resistance increase to prevent roof separation and damage; (b) the backfill body should have a certain amount of shrinkage to form a stable structure, with key blocks to provide support resistance; (c) the backfill body should have high later strength to meet the ventilation requirements of the second working face.(2)The uniaxial compressive strength of high-water materials is inversely proportional to the water-cement ratio, and it has good early setting characteristics and deformation capacity. Combined with the properties of high-water materials and the engineering conditions of the 2109 working face, the water-cement ratio was selected as 1.5:1. The key parameters of the backfill and support for gob-side entry retention were determined as the key parameters of filling and supporting the gob-side entry, which were 3.6 m for each filling length, 2.2 m for filling height and 1 m for filling width. These can provide a support strength of no less than 7.3 MPa, thereby guaranteeing gob-side entry retention will keep the surrounding rock stable.(3)We proposed a mine pressure-monitoring plan for gob-side entry retention at the 2109 working face, and analyzed the surrounding rock deformation and mine pressure appearance law of a gob-side entry retaining roadway. The displacement of the roof and floor of the roadway and the two sides, deformation of the backfill body, deformation of the roadway roof and stress on the anchor cable were all within a reasonable and controllable range, indicating that the high-water material backfill body had a good roadside support effect and met the engineering requirements.

## Figures and Tables

**Figure 1 ijerph-19-03913-f001:**
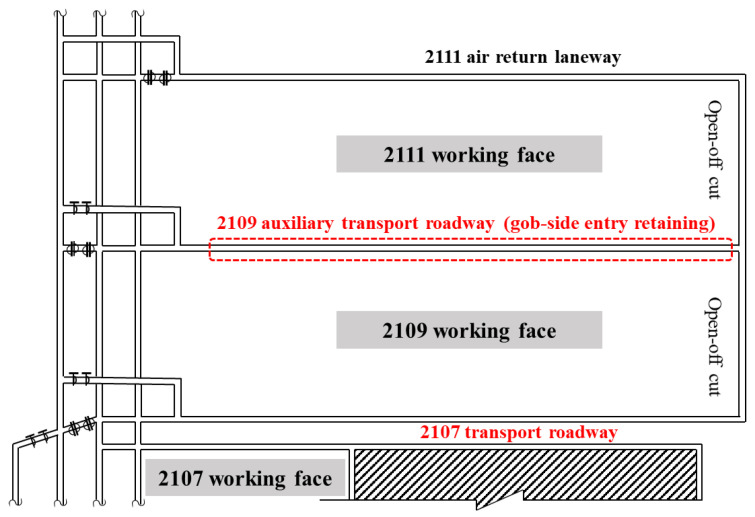
Schematic diagram of working surface layout.

**Figure 2 ijerph-19-03913-f002:**
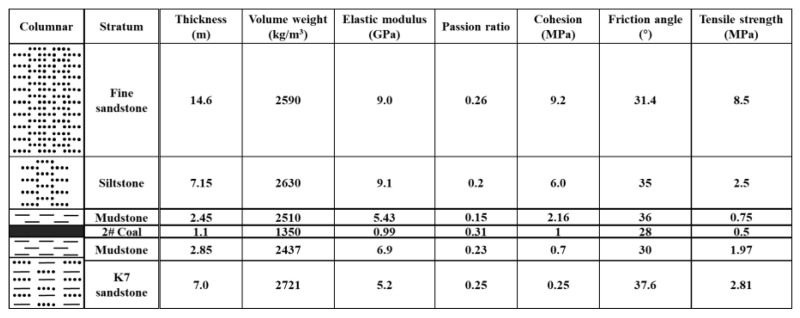
Stratigraphy of 2109 working face.

**Figure 3 ijerph-19-03913-f003:**
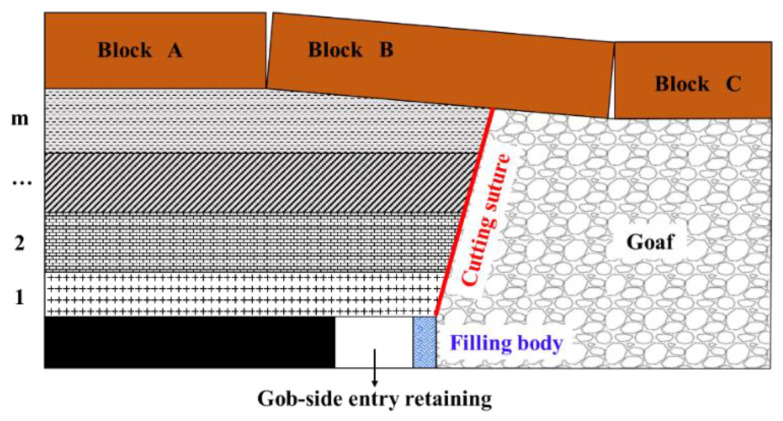
Schematic diagram of the roof cut to relieve pressure and stay along the goaf.

**Figure 4 ijerph-19-03913-f004:**
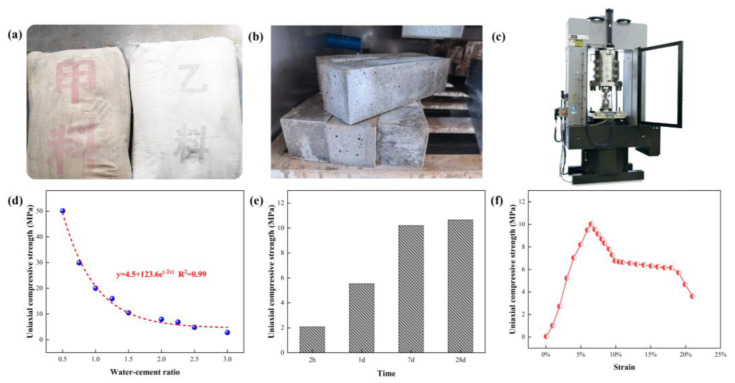
Preparation and performance of high-water filling body. (**a**) Raw high-water materials; (**b**) Prepared high-water filling body; (**c**) MTS 815 servo testing system; (**d**) Relationship between UCS and water-cement ratio; (**e**) UCS of high-water body at various ages; (**f**) Full stress-strain curve of high-water filling body).

**Figure 5 ijerph-19-03913-f005:**
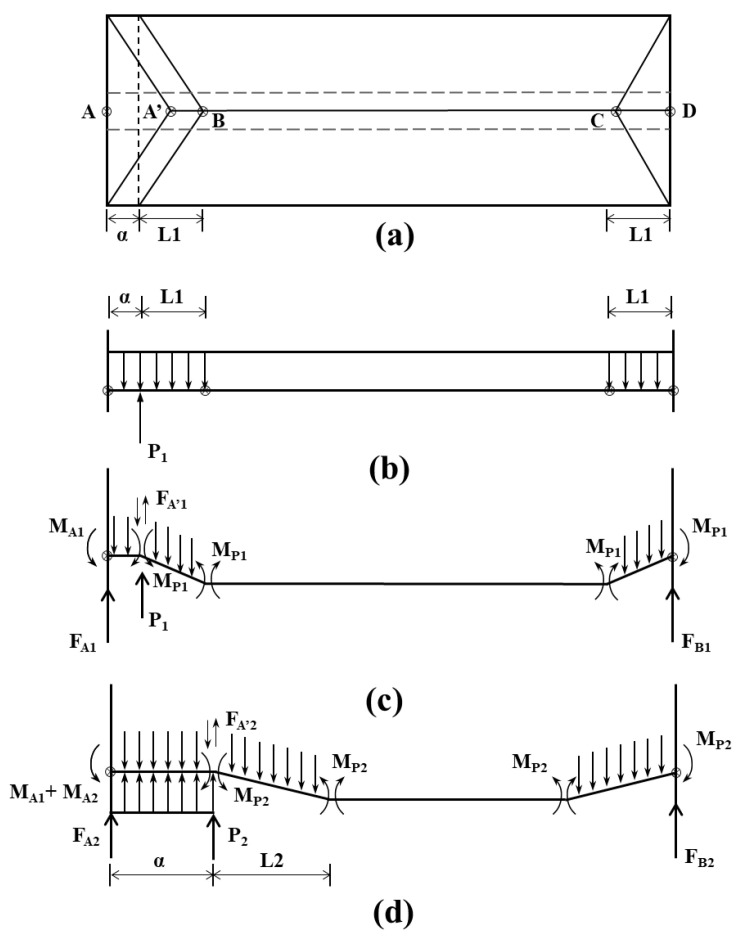
Calculation model of support resistance of gob-side entry retention. (**a**) Load strip partition method for rectangular top plate. (**b**) Long slats per unit width. (**c**) First layer support resistance of gob-side entry retention. (**d**) Second layer support resistance of gob-side entry retention.

**Figure 6 ijerph-19-03913-f006:**
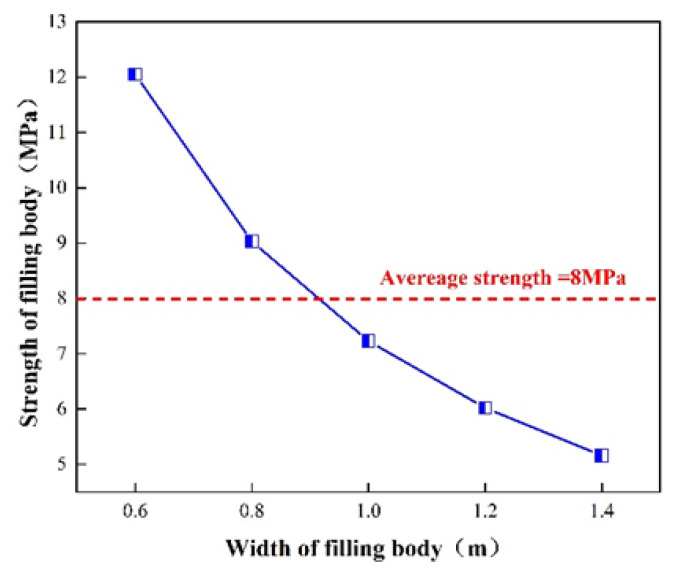
Relationship between strength and width of filling body.

**Figure 7 ijerph-19-03913-f007:**
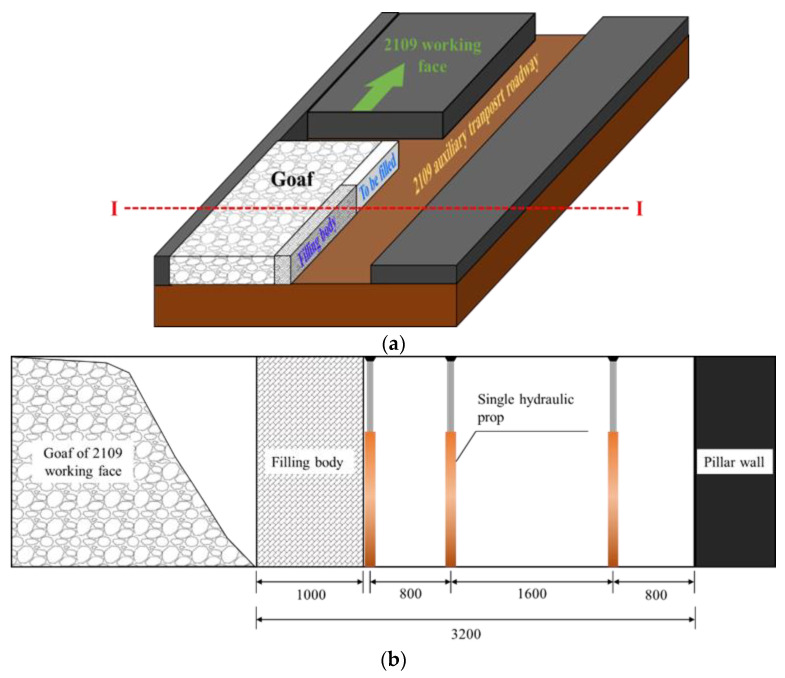
Schematic diagram of wall and roadway position section (A-A). (**a**) Stereoscopic view of the location of high-water materila filling body and roadway. (**b**) Layout of gangue retaining pillar.

**Figure 8 ijerph-19-03913-f008:**
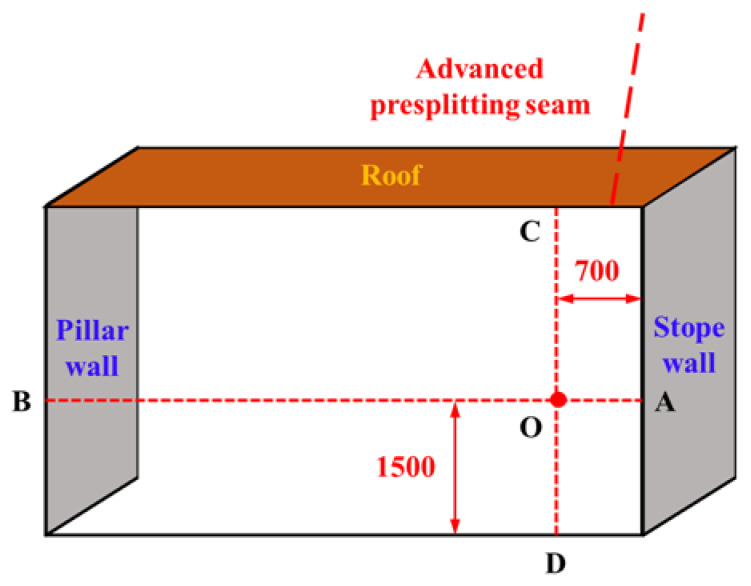
Diagram of layout of cross-observation.

**Figure 9 ijerph-19-03913-f009:**
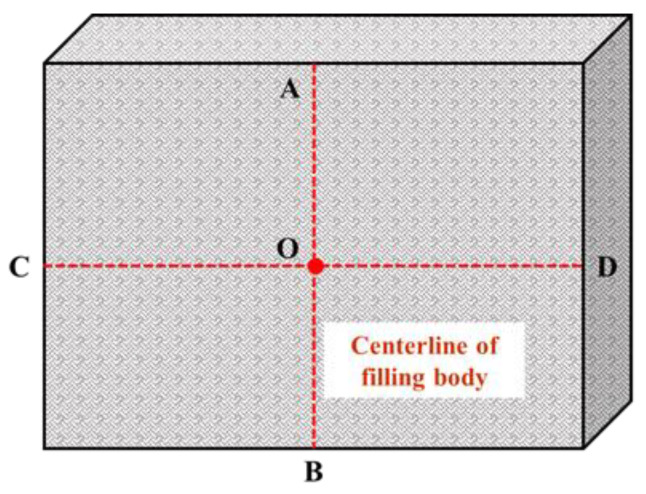
Schematic layout of surface displacement monitoring of filling wall.

**Figure 10 ijerph-19-03913-f010:**
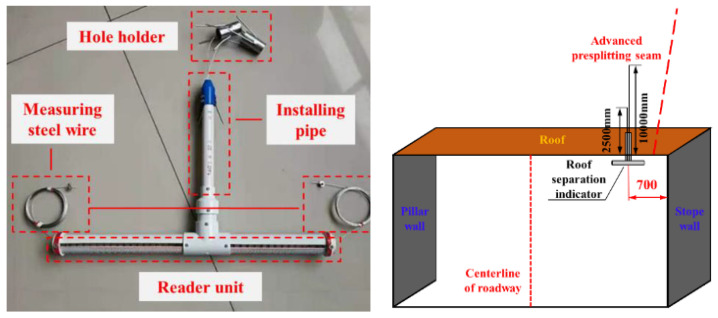
LBY-3 laminated instrument and its test layout.

**Figure 11 ijerph-19-03913-f011:**
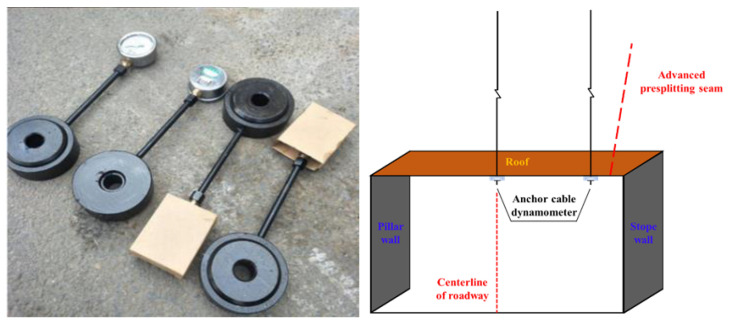
MCZ anchor cable dynamometer and its test layout.

**Figure 12 ijerph-19-03913-f012:**
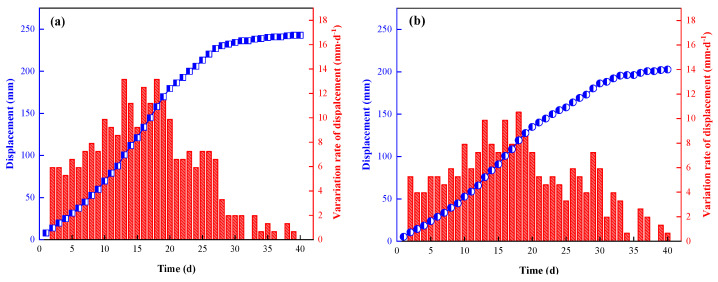
Variation law of displacement at measuring point No.1 ((**a**) top and floor plate; (**b**) two sides of roadway).

**Figure 13 ijerph-19-03913-f013:**
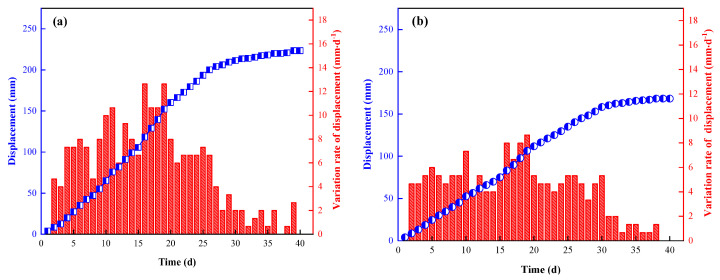
Variation law of displacement at measuring point No.2 ((**a**) top and floor plate; (**b**) two sides of roadway).

**Figure 14 ijerph-19-03913-f014:**
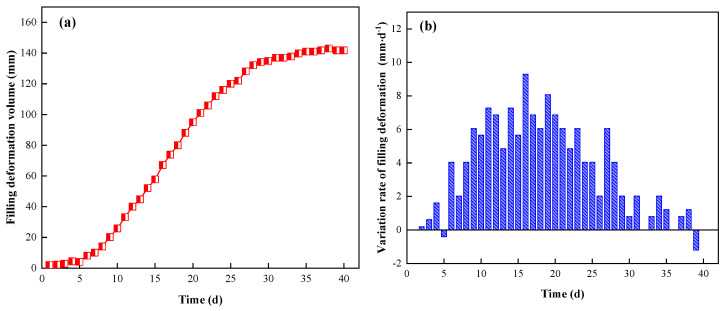
Variation law of filling body deformation ((**a**) deformation volume; (**b**) variation rate of filling body deformation).

**Figure 15 ijerph-19-03913-f015:**
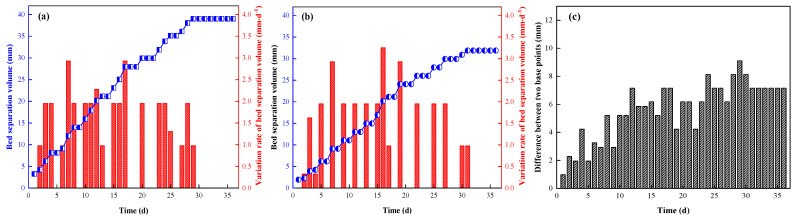
Variation law of bed separation at measuring point No.1 ((**a**) deep base point; (**b**) shallow base point; (**c**) difference between two base points).

**Figure 16 ijerph-19-03913-f016:**
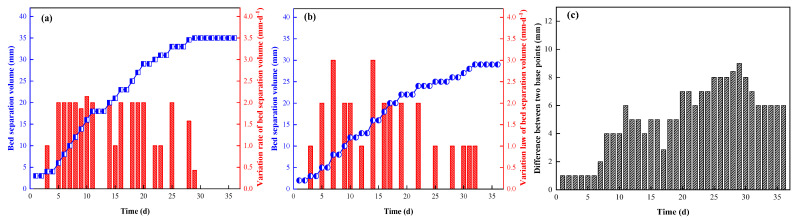
Variation law of bed separation at measuring point No.2 ((**a**) deep base point; (**b**) shallow base point; (**c**) difference between two base points).

**Figure 17 ijerph-19-03913-f017:**
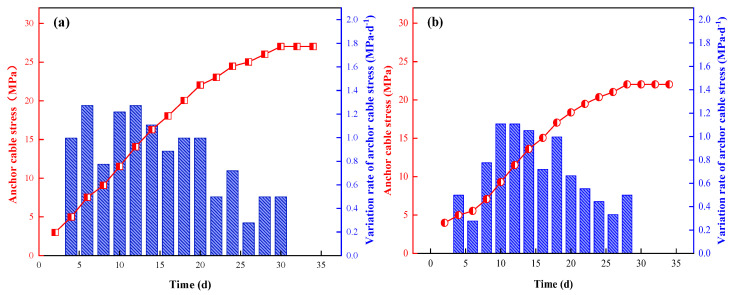
Variation of anchor cable stress at measuring point No.1 ((**a**) middle of roadway; (**b**) side of goaf).

**Figure 18 ijerph-19-03913-f018:**
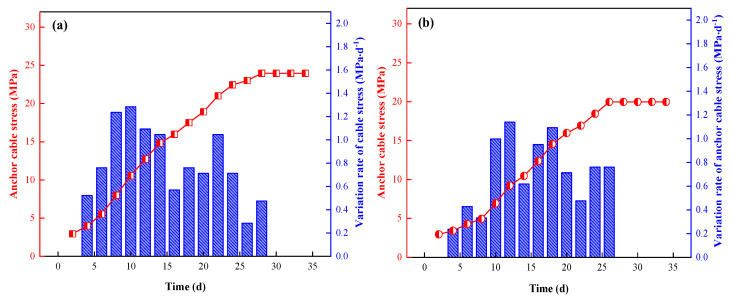
Variation of anchor cable stress at measuring point No.2 ((**a**) middle of roadway; (**b**) side of goaf).

**Table 1 ijerph-19-03913-t001:** Geological parameters of 2109 working face.

Parameter	*R*_t_ (MPa)	*γ* (KN/m^3^)	*a* (m)	*α* (°)	*h* (m)	*m*	*L*_*i*_ (m)
Value	8	25	4.5	30	2	5	20

## Data Availability

Not applicable.

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
