# Peer review of "Study on Filling Support Design and Ground Pressure Monitoring Scheme for Gob-Side Entry Retention by Roof Cutting and Pressure Relief in High-Gas Thin Coal Seam"

_ijerph, 2022, doi:10.3390/ijerph19073913_

Round 1

Reviewer 1 Report

The research work done and discussed in this manuscript is quite interesting.

The authors have done very good work, and discuss the results in detail.

However, the English language and style are not very good. Though interesting, I found it very hard to read several parts of the manuscript and to comprehend the scientific discussions made. This manuscript needs to be re-written almost entirely focusing on using shorter, simpler sentences and pay special attention to the grammar and ways of expression.

In addition, the introductory part needs a better conclusion in which the novelty of this work and the significance of the contribution of the authors are highlighted in a clear manner. Please think as readers and explain what the novelty of this work is, with respect to the existing literature.

In the methodology part, the text should be restructured and smaller sentences must be used.

Please also define all the parameters that are used in the mathematical equations, with reference to their units as well.

The research results are sufficient and sound.

This work is quite interesting but the way it is presented needs radical changes.

For further technical comments please see the annotated copy of the manuscript.

Author Response

Dear Reviewer, 

Thank you very much for your careful review. We have revised the manuscript according to your suggestion and marked it in red. Please check it. Once again, Thank you very much for your time and patience.

Best regards

Jifa Qian

Reviewer 2 Report

The manuscript „Study on filling support design and ground pressure monitoring scheme of gob side entry retaining by roof cutting and pressure relief in high gas thin coal seam” prepared by Hui Li and coauthors presents an important technical problem that must be solved during exploitation of one type of the coal deposits. Otherwise the technologic process may cause disturbances in the mine work or even a danger for the workers. The manuscript describes the solution taking into account rather practical than theoretical aspect.

The introduction and presentations of the problem, methods and results are prepared properly. The proposed method is realistic and should not cause the increase of the exploitation costs above a reasonable level. Thus, I think that the publication may be useful and this is its most valuable feature.

Nevertheless, I would suggest some improvements. The style of the text is “heavy”, with frequent repeating of the same words and slow explanation of the discussed problem. This has poor effect during reading, even causing an irksome feeling. I noted some linguistic errors, still easy for correction. Also title is very expanded, almost to the size of an abstract. A short and well elaborated title favors the remembering of the paper.

Detailed remarks are added to the manuscript as sticky notes.

I would suggest publication of this manuscript after moderate improvement.

Author Response

Dear Reviewer, 

Thank you very much for your valuable advice. We have revised the manuscript according to your suggestion and marked it in blue. We appreciate your warm work earnestly, and hope that the correction will meet with approval. 

Best regards

Jifa Qian

Round 2

Reviewer 1 Report

Thank you for taking into consideration the remarks made and for the additional effort in improving the soundness of the manuscript.